# Brief Mental Health Disorder Screening Questionnaires and Use with Public Safety Personnel: A Review

**DOI:** 10.3390/ijerph18073743

**Published:** 2021-04-03

**Authors:** Robyn E. Shields, Stephanie Korol, R. Nicholas Carleton, Megan McElheran, Andrea M. Stelnicki, Dianne Groll, Gregory S. Anderson

**Affiliations:** 1Anxiety and Illness Behaviours Laboratory, Department of Psychology, University of Regina, Regina, SK S4S 0A2, Canada; res996@uregina.ca (R.E.S.); korol20s@uregina.ca (S.K.); nick.carleton@uregina.ca (R.N.C.); andrea.stelnicki@uregina.ca (A.M.S.); 2Wayfound Mental Health Group, Calgary, AB T2R 1J5, Canada; meganM@wayfound.ca; 3Departments of Psychiatry and Psychology, Queen’s University, Kingston, ON N2L 3C5, Canada; grolld@queensu.ca; 4Faculty of Science, Thompson Rivers University, Kamloops, BC V2C 0C8, Canada

**Keywords:** brief screening tools, public safety personnel, mental disorders, questionnaires, MDD, GAD, panic disorder, PTSD, alcohol use disorder

## Abstract

Brief mental health disorder screening questionnaires (SQs) are used by psychiatrists, physicians, researchers, psychologists, and other mental health professionals and may provide an efficient method to guide clinicians to query symptom areas requiring further assessment. For example, annual screening has been used to help identify military personnel who may need help. Nearly half (44.5%) of Canadian public safety personnel (PSP) screen positive for one or more mental health disorder(s); as such, regular mental health screenings for PSP may be a valuable way to support mental health. The following review was conducted to (1) identify existing brief mental health disorder SQs; (2) review empirical evidence of the validity of identified SQs; (3) identify SQs validated within PSP populations; and (4) recommend appropriately validated brief screening questionnaires for five common mental health disorders (i.e., generalized anxiety disorder (GAD), major depressive depression (MDD), panic disorder, posttraumatic stress disorder, alcohol use disorder). After reviewing the psychometric properties of the identified brief screening questionnaires, we recommend the following four brief screening tools for use with PSP: the Patient Health Questionnaire-4 (screening for MDD and GAD), the Brief Panic Disorder Symptom Screen—Self-Report, the Short-Form Posttraumatic Checklist-5, and the Alcohol Use Disorders Identification Test-Consumption.

## 1. Introduction

A recent study by Carleton et al. [1] evidenced significant mental health disorder symptoms among public safety personnel (PSP). PSP include, but are not limited to, border services officers, correctional workers, firefighters, paramedics, police officers, and public safety communicators [2]. Nearly half (44.5%) of Canadian PSP screened positive for one or more mental health disorder(s) [1], a rate more than four times that of the general population (10.1%) [3]. Five mental health disorder symptom profiles were the most concerning, as rates surpassed those found in the general Canadian population: symptoms of alcohol use disorder (5.9% of PSP screened positive), anxiety (18.6% of PSP screened positive), depression (26.4% of PSP screened positive), panic disorder (8.9% of PSP screened positive), and posttraumatic stress disorder (23.2% of PSP screened positive) [1]. Such high rates of mental health disorder symptoms in PSP could be related to their frequent exposures to potentially psychologically traumatic events (PPTE) as part of their job duties [4].

Early identification of PSP with distressing symptoms could lead to earlier identification of mental health disorders [5,6,7], faster access to treatment, and better outcomes [8]. Screening questionnaires are a common way to identify those with mental health disorder symptoms. Screening questionnaires are generally brief, focused on only one or a few conditions, and can be self-administered, administered by a clinician, or administered by a specially trained assistant [9]. Screening questionnaires cannot provide a definitive diagnosis of a mental health disorder but are designed to help clinicians more efficiently identify symptoms indicative of mental health disorders requiring further assessment [9] and to help increase diagnostic consistency [10]. Third-party payers (e.g., Worker’s Compensation, insurance companies) may also require evidence of symptom changes as part of progress reporting obligations. Screening tools may be particularly beneficial for supporting rapid differential diagnostic efforts [6] for disorders such as alcohol use disorder (AUD) [5], major depressive disorder (MDD) [6], and posttraumatic stress disorder (PTSD) [7].

Regular screening may help PSP more quickly access care, but has the potential to be time consuming and costly [11]. A full battery of tools used to measure common mental disorder symptoms in PSP includes the Alcohol Use Disorders Identification Test (AUDIT) [12], the Generalized Anxiety Disorder Scale 7 (GAD-7) [13], the Patient Health Questionnaire 9 (PHQ-9) [14], the Panic Disorder Severity Scale-Self-Report (PDSS-SR) [15], and the Posttraumatic Checklist for DSM-5 (PCL-5) [16]. The AUDIT is a 10-item scale that takes approximately 2 to 4 min to complete [17]. The GAD-7 has 7 items and takes approximately 3 min to complete [18]. The PHQ-9 is a 9-item scale that takes approximately 3 min to complete [19]. The PDSS-SR is a 7-item scale that takes approximately 3 min to complete (based on the length of time it takes to complete the PHQ-9 and GAD-7, we estimate ~3 min to complete) and the PCL-5 is a 20-item scale that takes approximately 5–10 min to complete [20]. In total, a full battery of tools could take up to 20 min to complete. Respondents may also perceive longer self-report measures as a barrier to screening, especially if given five measures to complete at one time. In any case, a recent review of mandatory psychological screening suggests that benefits from frequent screening include reduced stigma and barriers for accessing mental health care; self-identifying mental health concerns; and identifying and highlighting available mental health treatments and services [21]. To our knowledge, there has yet to be a literature review identifying validated, reliable, very short screening tools with useful sensitivity and specificity for PSP.

### Objectives

The current study was designed to address the following objectives: (1) identify existing brief mental health disorder screening questionnaires; (2) review empirical evidence of the validity of identified screening questionnaires; (3) identify screening questionnaires validated within PSP populations; and (4) provide recommendations for appropriate and validated brief screening questionnaires for PSP.

## 2. Materials and Methods

The current literature review was conducted between September and December 2019 and was designed to identify brief mental health disorder screening questionnaires that may be used to support PSP. The focus was on identifying validated questionnaires that screen for one or more mental disorders identified as prominent for PSP (i.e., depression, anxiety, PTSD, panic disorder, and alcohol use disorder) [1] using the lowest possible number of items (e.g., PHQ-4, which includes two items related to major depressive disorder and two items related to generalized anxiety disorder, rather than the PHQ-9 and GAD-7).

### 2.1. Search Strategy

The review was based on search results from several databases—specifically, PsycARTICLES (Ovid), PsycInfo (Ovid), Web of Science, and Google Scholar. An initial search was conducted using the following search terms: brief psychology screening instruments, mental disorder screening, short form of mental disorder measures, psychological symptom questionnaires, short symptom measures, depression symptom measures, anxiety symptom measures, alcohol symptom measures, substance use symptom measures, and post-trauma symptom measures. Variations and combinations of the search terms were used to follow up on preliminary searches and identify additional questionnaires. Publications written in a language other than English, as well as dissertations and books in any language, were excluded. Brief tools that did not screen for at least one of the five major disorders (i.e., alcohol use disorder, anxiety, depression, panic disorder, and PTSD) were also not included. Validation papers identified for each brief screening questionnaire were located using Google Scholar. A list of brief screening questionnaires that were validated in clinical or general populations was then compiled.

A subsequent search was conducted to explore the use of the identified questionnaires among PSP. Databases used for the subsequent search included PsycInfo (Ovid) and Google Scholar. Abbreviations for previously identified brief screening questionnaires were used as PsycInfo search terms. The abbreviations included were as follows: 90 to 120 Day Post-Deployment Psychological Short Screen, ADD, ASSIST, CID-S, CIDI-SC, MHI-5, K6, MCS-12, M.I.N.I., M-3, PHQ-4, PAS, WSQ, AUDIT-C, Brief PSS, PADIS, PHQ-2, PC-PTSD-5, PCL-C-SF, and SPRINT. PSP job titles (i.e., police, firefighter, paramedic, corrections officer, public safety personnel, and first responder) were paired with each of the brief screening questionnaire abbreviations in order to identify studies using the screening questionnaires with PSP participants. Finally, a search was conducted to identify associated extant research specifically validating each tool for use with PSP.

### 2.2. Data Analysis

Psychometric properties for each questionnaire were reported in terms of sensitivity (i.e., proportion of true cases detected) and specificity (i.e., proportion of true non-cases correctly classified as non-cases) of screening questionnaires, as well as positive likelihood ratios (i.e., the odds that a positive test result would be expected in a patient with the associated diagnosis), positive predictive values (PPV, i.e., percentage of positive screens that are accurate), negative predictive values (NPV, i.e., percentage of negative screens that are accurate), and receiver operating characteristic (ROC) curves. ROC curves indicate sensitivity and specificity of possible cut-off scores, and the area under the ROC curve (AUC) indicates the diagnostic power of the questionnaire in discriminating between those with and without the diagnosis of interest [22]. AUC statistics range from 0.5 and 1.0. A value of 0.5 indicates that the questionnaire is performing at a chance level in discriminating between those with and without the diagnosis of interest and 1.0 indicates perfect discrimination. Swets [23] has suggested that values between 0.50 and 0.70 represent low accuracy, 0.70 and 0.90 indicate a useful screening questionnaire, and 0.90 or higher indicate a highly accurate screening questionnaire. In addition, internal consistency (i.e., Cronbach’s α) and interrater reliability of the questionnaires were reported when available. The following sections first review brief screening questionnaires for multiple symptoms, reporting on available validation data among PSP, followed by brief screening questionnaires for single-disorder symptoms, again noting if the measure has been validated in a PSP population.

## 3. Results

### 3.1. Multiple Symptom Screening Questionnaires

#### 3.1.1. The 90 to 120 Day Post-Deployment Psychological Short Screen

The United States Army Medical Research Unit-Europe developed a screening questionnaire to provide soldiers with an easy means to self-identify mental health issues and receive counseling within 90 to 120 days post-deployment. The 90 to 120 Day Post-Deployment Psychological Short Screen [24] is a 15-item screening questionnaire for five areas: traumatic stress (4 items), depression (4 items), relationship problems (2 items), alcohol problems (2 items), and anger problems (3 items). Researchers assessed the criterion validity of the 90 to 120 Day Post-Deployment Psychological Short Screen in comparison to the Mini-International Neuropsychiatric Interview (M.I.N.I.) [10]. For the traumatic stress items, a cut-off value of ≥3 positive responses was recommended (sensitivity = 0.46; specificity = 0.97). For the relationship problems items, a positive score on both items was recommended (sensitivity = 0.40 to 0.82; specificity = 0.89 to 0.92). For the anger problems items, soldiers who endorse “sometimes,” “often,” or “very often” on any two of the three items was recommended as a cut-off (sensitivity = 0.59; specificity = 0.91). For the depression items, a cut-off value of ≥ 2 positive responses was recommended (sensitivity = 0.50 to 0.53; specificity = 0.94 to 0.96). For the alcohol problems items, a positive score on both items was recommended (sensitivity = 0.33; specificity = 0.94; see Table 1). The 90 to 120 Day Post-Deployment Psychological Short Screen appears to be a psychometrically-sound screening questionnaire for traumatic stress, relationship problems, anger problems, depression, and alcohol problems [24].

#### 3.1.2. Anxiety and Depression Detector (ADD) 

The ADD is a 5-item self-report screening questionnaire for identifying primary care patients with anxiety and depression [25]. The ADD is used to screen for symptoms of panic disorder (PD), PTSD, social phobia (SP), generalized anxiety disorder (GAD), and major depressive disorder (MDD) over the past three months. The ADD has been tested with approximately 800 primary care patients [25]. Split-half item analyses of a randomly selected half of the sample were conducted to narrow down better-performing items in the initial questionnaire. Split-half item sensitivity ranged from 0.49 to 1.00 and specificity from 0.48 to 0.81 when compared to a computerized version of the Composite International Diagnostic Interview (CIDI) [44] assessing for Diagnostic and Statistical Manual of Mental Disorders 4th edition (DSM-IV) [45]. Cross-validation of the better-performing items in the second half of the sample resulted in sensitivities ranging from 0.62 to 1.00 and specificities ranging from 0.56 to 0.83 (see Table 1). When a “Yes” answer to any of the screening questions was used in predicting any diagnosis, the sensitivities ranged from 0.92 to 0.96 across the two versions and specificities ranged from 0.57 to 0.82. The ADD appears to be a psychometrically-sound screening questionnaire for PD, PTSD, SP, GAD, and MDD among primary care patients [25]. The conducted search did not identify any studies in which the ADD was used specifically with PSP populations.

#### 3.1.3. Alcohol, Smoking, and Substance Involvement Screening Test-Version 2 (ASSIST-2)

The ASSIST-2 is an 8-item self-report screening questionnaire that investigates frequency of use and associated problems for tobacco, alcohol, cannabis, cocaine, amphetamine-type stimulants (ATS), inhalants, sedatives, hallucinogens, opioids, and other drugs in the past three months [26]. Researchers demonstrated concurrent validity of the ASSIST-2 with measures of addiction (*r* = 0.76 to 0.88 for alcohol, cannabis, cocaine, amphetamines, sedatives, and opioids), dependence (*r* = 0.59), disordered alcohol use (*r* = 0.82), and tolerance (*r* = 0.78) [26]. Construct validity was established by assessing for significant correlations between ASSIST-2 scores and measures of risk factors for the development of substance use problems (*r* = 0.48 to 0.76). In addition, the ASSIST-2 items assessing dependence and abuse were significantly correlated with M.I.N.I. [10] scores for dependence (*r* = 0.76) and abuse (*r* = 0.75) [26].

Discriminant validity for the ASSIST-2 was evaluated by assessing the effectiveness for significantly discriminating between substance use, abuse, and dependence (*p* ≤ 0.001) [26]. ROC analyses were also used to establish cut-off scores with suitable specificities (0.50 for sedative abuse/dependency, to 0.96 for opioid use/abuse) and sensitivities (0.54 for sedative abuse/dependency to 0.97 for amphetamine use/abuse; see Table 1). Cut-off scores are as follows: total substance involvement (use/abuse = 14.5; abuse/dependence = 28.5), alcohol (use/abuse = 5.5.; abuse/dependence = 10.5), cannabis (use/abuse = 1.5; abuse/dependence = 10.5), cocaine (use/abuse = 0.5; abuse/dependence = 8.5), amphetamines (use/abuse = 0.5; abuse/dependence = 11.5), sedatives (use/abuse = 9.5; abuse/dependence = 10.5), opioids (use/abuse = 0.5; abuse/dependence = 14.5) [26]. The ASSIST-2 appears to have similar specificity, sensitivity, and AUC in several languages (e.g., French, Spanish) [46,47], for different ages (e.g., adolescents) [48], and for diverse cultures (e.g., Pacific People of New Zealand) [49]. The ASSIST-2 appears to be a psychometrically-sound screening questionnaire for identifying substance use among individuals who use several substances in varying degrees [26]. The conducted search did not identify any studies in which the ASSIST-2 was specifically used with PSP populations.

#### 3.1.4. Composite International Diagnostic-Screener (CID-S)

The CID-S is a 12-item self-report screening questionnaire for somatoform, anxiety, depressive and other affective, and substance use disorders (SUDs). The CID-S was developed based on the core diagnostic questions from the full CIDI [44] for the assessment of DSM-IV [45] and International Classification of Diseases (ICD-10) [50] diagnoses [27]. Researchers tested the CID-S in comparison to the full-scale version using a random sample of 1095 participants. Overall sensitivity for the CID-S was 0.85 (0.79 for alcohol use/dependence to 1.00 for PD; see Table 1), and the overall NPV was 0.92 (range = 0.96 for SD to, 1.00 for mania) for any current diagnosis (12-month DSM-IV diagnosis). The CID-S appears to be a psychometrically-sound screening questionnaire for identifying DSM-IV [45] and ICD-10 [50] diagnoses [27]. The conducted search did not identify any studies in which the CID-S was specifically used with PSP populations.

#### 3.1.5. Composite International Diagnostic Interview Screening Scale (CIDI-SC)

The CIDI-SC is a 9- to 38-item screening questionnaire for DSM-IV major depressive episode (MDE), GAD, PD, and bipolar disorder. There are nine core questions and up to 38 follow-up questions, depending on responses. Researchers have assessed the CIDI-SC for diagnostic accuracy based on the Structured Clinical Interview for DSM-IV Diagnoses (SCID-IV) [51]. Results demonstrated excellent concordance (based on AUC) between CIDI-SC and SCID-IV diagnoses for MDE (0.93), GAD (0.88), PD (0.90), and bipolar disorder (0.97). In addition, PPV ranged from 48.2% (GAD) to 73.7% (bipolar disorder). Sensitivity ranged from 0.68 to 0.80 and specificity from 0.90 to 0.99 (see Table 1) [51]. The CIDI-SC appears to be a psychometrically-sound screening questionnaire for mood and anxiety disorders [28]. The conducted search did not identify any studies in which the CIDI-SC was specifically used with PSP populations.

#### 3.1.6. Five-Item Mental Health Inventory (MHI-5)

The MHI-5 [29] is a 5-item self-report screening questionnaire for affective and anxiety disorders developed from the original 38-item version [52]. There are three items measuring depressive symptoms and psychological well-being, and two items measuring anxiety symptoms [29]. Researchers compared accuracy of the MHI-5 with the measures of mental health, somatic symptoms, and general health [29]. The MHI-5 detected depressive disorders (AUC = 0.78), anxiety disorders (AUC = 0.74), and any diagnosable disorder (AUC = 0.79) [29]. Other researchers [30] also compared the diagnostic accuracy of the MHI-5 with a computerized version of the CIDI [53] in detecting past month diagnoses based on the DSM-III-R [54]. Different cut-off scores were recommended depending on the use of the measures [30]. There have been three cut-off scores calculated: (1) the maximum sum of sensitivity and specificity; (2) the costs of a false negative are fifty times higher than the costs of a false positive; and (3) the costs of a false negative are five times higher than the costs of a false positive [30]. The cut-off score for major depression and/or dysthymia using the first two methods was 74, but was 54 using the third method [30]; therefore, attention should be paid to scoring method when using or reviewing results from the MHI-5.

In adults, the MHI-5 AUC scores have been high for mood/affective disorders (any = 0.88; MDD = 0.92; dysthymia = 0.91; bipolar disorder = 0.88), anxiety disorders (any = 0.71; GAD = 0.90; PD = 0.87; agoraphobia = 0.80; SP = 0.77; simple phobia = 0.69; obsessive compulsive disorder (OCD) = 0.93; see Table 1) [30,31]. Researchers have found evidence that the sensitivity and specificity of the MHI-5 in comparison with SCID-IV [51] diagnoses was inadequate for somatoform and SUDs (AUC of 0.65 to 0.66), especially in cases without comorbid anxiety or mood disorder diagnoses [31]. The MHI-5 appears to be a psychometrically-sound screening questionnaire for mood and anxiety disorders [30]. The conducted search did not identify any studies in which the MHI-5 was specifically used with PSP populations.

#### 3.1.7. Kessler Psychological Distress K6 Screening Scale (K6)

The K6 is a 6-item self-report screening questionnaire for serious mental illness (SMI) based on the DSM-IV [45]. The K6 was created based on general population analyses in 14 countries with the WHO Mental Health Survey Initiatives to estimate SMI population prevalence [32]. SMI is defined as an individual experiencing at least one 12-month DSM disorder (excluding substance use disorder) that causes “serious impairment” [32]. The K6 requires respondents to rate how often they felt “nervous,” “hopeless,” “restless or fidgety,” “so depressed that nothing could cheer you up,” “that everything was an effort,” and “worthless over the past month and in the worst month in the past 12 months.” The K6 has good concordance with clinical diagnoses of SMI in a random sample of the general United States population [55,56]. The K6 can detect moderate SMI with a cut-off score greater than five (AUC = 0.82, specificity = 0.75, sensitivity = 0.760 [57]. The K6 AUC for identifying individuals with SMI in the past month ranges from 0.76 to 0.89 across a nationally representative World Mental Health survey of 14 countries (i.e., Nigeria, South Africa, Brazil, Colombia, Mexico, United States, India, Japan, New Zealand, Beijing, Shanghai, Shenzhen, Bulgaria, Romania, Ukraine, Lebanon; see Table 1) [32]. The K6 performs as well as a longer 10-item version [32] and has been translated and validated for Arabic [58] and Korean [59]. The K6 appears to be a psychometrically-sound screening questionnaire for SMI, but is not designed for specific diagnoses [32]. The K6 has been used with PSP populations (i.e., Canadian police officers and volunteer Australian firefighters; see Table 1) [33,34]; however, the psychometric properties were not well reported in these studies. Janzen et al. [33] reported only internal consistency (Cronbach’s α = 0.81) and Milligan-Saville et al. [34] did not report psychometric statistics.

#### 3.1.8. Mental Health Component Scale (MCS-12)

The MCS-12 [35] is a 12-item self-report screening questionnaire based on the 12-item Short-Form Health Survey [60]. The MCS-12 is used to screen for depression and anxiety disorders in the general population. Specifically, the MCS-12 assesses self-reported psychological symptoms and limitations due to mental health over the past 4 weeks [61]. The MCS-12 sensitivity for depression, anxiety disorders (i.e., PD, agoraphobia, SP, GAD, OCD, PTSD), and any common mental disorder was tested relative to other brief screening questionnaires [35] and using diagnoses based on the CIDI [44]. The MCS-12 AUC scores were good for depression (0.92), anxiety disorders (0.83), and any common mental disorder (0.86; see Table 1) [35]. The cut-off scores for detecting depression or any anxiety disorder are reported in Table 1 [35]. The MCS-12 appears to be a psychometrically-sound screening questionnaire for mood and anxiety disorders [35]. The conducted search did not identify any studies in which the MCS-12 was specifically used with PSP populations.

#### 3.1.9. Mini-International Neuropsychiatric Interview (M.I.N.I.) Screen

The M.I.N.I. Screen [10] is a 21-item dimensional self-report screening questionnaire for MDD (2- items), dysthymia (1-item), suicidality (1-item), mania (2 items), PD (2- items), SP (1-item), OCD (2 items), PTSD (3 items), alcohol abuse (1-item), substance abuse (1-item), eating disorders (4 items), and GAD (1-item), based on DSM-IV [45] and ICD-10 [50] criteria. The M.I.N.I. Screen is a subset of M.I.N.I. items and does not appear to be used without the full M.I.N.I., which has substantial support as a diagnostic tool [10,62]. The clinician-rated M.I.N.I. sensitivity scores range from 0.45 (current drug dependence) to 0.96 (MDD), with PPV from 43% (current simple phobia) to 90% (anorexia; see Table 1). The patient-rated M.I.N.I. sensitivity scores range from 0.39 (current drug dependence) to 0.89 (alcohol dependence), with PPV from 11% (dysthymia) to 75% (MDD and anorexia) [10]. Concordance between the clinician-rated M.I.N.I. and SCID diagnoses ranged from ϰ values of 0.43 (current drug dependence) to 0.90 (anorexia nervosa) [10]. Concordance between the clinician-rated M.I.N.I. and CIDI diagnoses ranged from ϰ values of 0.36 (GAD) to 0.82 (alcohol dependence) [10]. Interrater reliability of the clinician-rated M.I.N.I. ranged from ϰ values of 0.79 (current mania) to 1.00 (MDD, OCD, current alcohol dependence, eating disorders; see Table 1) [10]. Test–retest reliability of the clinician-rated M.I.N.I. ranged from ϰ values of 0.35 (current mania) to 1.00 (bulimia; see Table 1) [10]. The M.I.N.I. appears to be a psychometrically-sound questionnaire for assessing DSM-IV [45] and ICD-10 [50] diagnostic criteria in less time than the SCID-III-R or CIDI [10]. The M.I.N.I. has been used as a diagnostic screening questionnaire in a study testing the efficacy of a cognitive behavioural intervention among PSP; however, psychometric properties of the M.I.N.I in this sample were not reported (see Table 1) [36].

#### 3.1.10. My Mood Monitor (M-3) Checklist

The M-3 Checklist is 27-item self-report screening questionnaire for current (i.e., past 2 weeks) MDD (7 items), GAD (2 items), PD (2 items), social anxiety disorder (SAD; 1-item), OCD (3 items), PTSD (4 items), lifetime bipolar disorder (4 items), and general functional impairment (4 items) [37]. The M-3 Checklist sensitivity ranges from 0.82 (anxiety module) to 0.88 (bipolar disorder and PTSD), and specificity ranged from 0.70 (bipolar disorder) to 0.80 (MDD) when compared to the M.I.N.I. (see Table 1) [37]. The M-3 Checklist appears to be a psychometrically-sound screening questionnaire for multiple diagnoses and takes less than 5 min to complete [37]. The conducted search did not identify any studies in which the M-3 Checklist was specifically used with PSP populations.

#### 3.1.11. Patient Health Questionnaire-4 (PHQ-4)

The PHQ-4 [39] is a 4-item self-report screening questionnaire for depression and anxiety symptoms experienced in the past 2 weeks, consisting of the PHQ-2 (2 core criteria for depression) [63] and the GAD-2 (2 core criteria for GAD) [64]. Construct validity of severity scores of the PHQ-4 was measured in comparison to mean number of disability days and physician visits, and demonstrated a strong incremental relationship [39,65]. The PHQ-2 AUC score for any anxiety disorder is 0.78 and ranges from 0.75 (PD), to 0.81 (GAD) and the GAD-2 AUC scores for any anxiety disorder is 0.84 and ranges from to 0.80 (PTSD), to 0.91 (GAD) [39]. The total PHQ-4 AUC scores range from 0.79 (MDD) to 0.84 (GAD) [38]. Internal consistencies of the PHQ-4 (α = 0.81), PHQ-2 (α = 0.76), and GAD-2 (α = 0.84) were acceptable [38]. Cut-off scores of ≥ 3 for each of the PHQ-2 and GAD-2 scores of the PHQ-4 appear to identify probable cases of depression or anxiety [63,66]. The PHQ-4 appears to have good test–retest reliability (ϰ = 0.69–0.81) [38]. The PHQ-4 appears to be a psychometrically-sound screening questionnaire for mood and anxiety disorders [39,65]. The PHQ-4 has been used in a sample of police officers as part of an investigation of the relationship between job demands and the development of emotional exhaustion, depression, and anxiety; however, the only measure of reliability reported from the policing sample was Cronbach’s α [40].

#### 3.1.12. Posttraumatic Adjustment Scale (PAS)

The PAS is a 10-item self-report screening questionnaire for patients at high risk for PTSD and/or depression during hospitalization following a traumatic injury [41]. The PAS items were derived from predictor variables (e.g., prior trauma, emotional response during trauma, coping self-efficacy) with moderate to high effect sizes in the PTSD vulnerability literature [41]. The PAS has a 5-factor solution and each factor has 2 items (i.e., Factor 1 = Acute Stress Response; Factor 2 = Prior Social Support; Factor 3 = Prior Psychiatric/Trauma History; Factor 4 = Self-Efficacy; and Factor 5 = Perceived Threat/Pain). Using data from trauma centre patients, the PAS has a sensitivity of 0.82 and specificity of 0.84 for PTSD, and sensitivity of 0.72 and specificity of 0.75 for MDE (see Table 1) [41]. The risk for PTSD is calculated by summing all 10 items and a score of 16 or higher represents high risk for later development of PTSD [41]. The risk for MDE is calculated by summing 5 of the 10 items and a score of 4 or above on these items represents high risk for later development of MDE [41]. The PAS appears to be a psychometrically-sound screening questionnaire for risk of PTSD and MDE [41]. The conducted search did not identify any studies in which the PAS was specifically used with PSP populations.

#### 3.1.13. Web Screening Questionnaire (WSQ)

The Web Screening Questionnaire (WSQ) is a 15-item online dimensional self-report screening questionnaire for depressive disorder, alcohol abuse/dependence, GAD, PTSD, SP, PD, agoraphobia, specific phobia, and OCD. The WSQ sensitivity ranged from 0.72 (SP) to 1.00 (agoraphobia) and specificity from 0.44 (PD) to 0.77 (PD with agoraphobia; see Table 1) when using the CIDI [44] as a reference [42]. PPV ranged from 11% (PTSD) to 51% (any depressive disorder), while NPV ranged from 87% (specific phobia) to 100% (agoraphobia) [42]. The WSQ appears successful for identifying true negatives but may be susceptible to false positives [42]. Recent research [43] found specificity for depressive disorders, anxiety disorders, and alcohol abuse/dependence ranged from 0.89 to 0.97, with the exception of PTSD (0.52) and specific phobia (0.73) when using the M.I.N.I.-Plus [10] as a reference measure for DSM-IV-TR diagnoses [67]. Sensitivity ranged from 0.67 to 1.00, with the exception of depressive disorder (0.56) and alcohol abuse/dependence (0.56; see Table 1), and NPVs were high in the general population (greater than 87%), but PPVs were low (2–33%) [43]. The WSQ appears to be a psychometrically-sound screening questionnaire for depressive or anxiety disorders, but the WSQ is susceptible to many false positives [42,43]. The conducted search did not identify any studies in which the WSQ was specifically used with PSP populations.

### 3.2. Single-Disorder Symptom Questionnaires

#### 3.2.1. Alcohol Use Disorders Identification Test-Consumption (AUDIT-C)

The AUDIT-C is a 3-item questionnaire that was first validated for identification of alcohol misuse in Veterans Affairs (VA) outpatients [68]. Researchers validated the AUDIT-C in comparison to the full AUDIT [12] and other measures of problematic alcohol use [68]. The AUDIT-C performed better than the full AUDIT in terms of AUC (*p* = 0.03); however, the full AUDIT had a higher AUC for detecting alcohol abuse and/or dependence (*p* < 0.001) [68]. Researchers demonstrated an optimal scoring cut-off of ≥3, for a total of 12 points for identifying alcohol abuse/dependence (sensitivity = 0.90; specificity = 0.45; see Table 2) and heavy drinking (sensitivity = 0.98; specificity = 0.56) [68]. The AUDIT-C has also been validated within the three main ethnic demographic categories (African American [AA], Caucasian [C], and Hispanic [H]) in the United States with similar results (AA: sensitivity = 0.67–0.76; specificity = 0.92–0.93; AUC = 0.90–0.95; C: sensitivity = 0.70–0.95; specificity = 0.91–0.89; AUC = 0.86–0.95; H: sensitivity = 0.85; specificity = 0.84–0.88; AUC = 0.91–0.93) [69]. Further, the Korean language version of AUDIT-C has been validated with a sample of PSP [70]. Using a total cut-off score of 7.5 (sensitivity = 0.82; specificity = 0.80) to identify any alcohol use disorder and a cut-off score of 8.5 (sensitivity = 0.86; specificity = 0.86) to identify those with alcohol dependence, the Korean AUDIT-C appears to be a valid screening tool for PSP [70]. The AUDIT-C has also been used in numerous studies of PSP mental health, with internal consistency ranging from 0.67 to 0.81 when analyzed [71,72,73,74]. The AUDIT-C appears to be a psychometrically-sound screening questionnaire for alcohol abuse or dependence and heavy drinking [68].

#### 3.2.2. Brief Panic Disorder Severity Scale-Self Report (Brief PDSS-SR)

The PDSS-SR is a 2-item self-report measure for assessing panic disorder symptoms [75]. The items were chosen from the original Panic Disorder Severity Scale-Self Report [15] after examining factor structure, sensitivity to change, and clinical representativeness. The 2-item scale showed good psychometric properties (ω_C_ = 0.74) and at the optimal cut-off score of >3, AUC was 0.82 (sensitivity = 0.85; specificity = 0.66) [75]. The scale has not yet been validated in other languages or cultures. The conducted search did not identify any studies in which the Brief PSS was specifically used with PSP populations.

#### 3.2.3. Brief PTSD Screen (Brief PSS)

The Brief PSS [76] is a 3-item self-report screening questionnaire derived from the 17-item PTSD Symptom Scale (PSS) [84,85]. The Brief PSS is used to screen for PTSD in clients with SMI (e.g., schizophrenia, MDD, bipolar disorder). The Brief PSS appears to have good internal consistency (α = 0.81), correlates highly with the 17-item PSS (*r* = 0.93), and has a demonstrated AUC score of 0.96 [76]. A cut-off score of ≥3 can be used to screen for PTSD (sensitivity = 0.94; specificity = 0.85; see Table 2). The Brief PSS appears to be a psychometrically-sound brief screening questionnaire for PTSD among individuals with SMI [76]. The conducted search did not identify any studies in which the Brief PSS was specifically used with PSP populations.

#### 3.2.4. Panic Disorder Screener (PADIS)

The PADIS is a 4-item self-report screening questionnaire for PD [77]. The PADIS appears to have good internal consistency (α = 0.86) and was able to distinguish individuals with PD from those without [77] based on the M.I.N.I. [10]. A cut-off score of 4 or higher distinguished panic cases from non-cases (specificity = 0.84; sensitivity = 0.77; AUC = 0.87; see Table 2; 77). The PADIS had lower PPV (25%) but excellent NPV (98%). Every one-point increase in PADIS total score is associated with a 68.6% increase in the odds of meeting clinical criteria for PD [77]. The PADIS appears to be a psychometrically-sound screening questionnaire for PD [77]. The conducted search did not identify any studies in which the PADIS was specifically used with PSP populations.

#### 3.2.5. The Patient Health Questionnaire-2 (PHQ-2)

The PHQ-2 [63] is a self-report screening questionnaire for MDD consisting of the first two items of the PHQ-9 [14]. The PHQ-2 is used to screen for the core symptoms of MDD over the past two weeks. PHQ-2 scores of ≥3 as a cut-off result in sensitivity of 0.83 and specificity of 0.92 for detecting MDD (see Table 2) [63]. Researchers identified PHQ-2 cut-off score of 3 as the optimal cut-point for screening for MDD [63]; however, a recent meta-analysis of 21 validation studies suggested a cut-off score of ≥2 may be better, resulting in pooled sensitivity of 0.91 and specificity of 0.70, compared to pooled sensitivity of 0.76 and specificity of 0.87 for a cut-off score of ≥3 [86]. The PHQ-2 has been used in one study investigating the relationships between police officer wellness and safety; however, no psychometric properties or validation of the PHQ-2 were reported [73]. The PHQ-2 appears to be a psychometrically-sound screening questionnaire for MDD [63,66].

#### 3.2.6. Primary Care Posttraumatic Stress Disorder Screen (PC-PTSD-5)

The PC-PTSD-5 is a 4-item primary care screening questionnaire that has been updated from the original version (PC-PTSD) [87] to reflect DSM-5 criteria [88] for PTSD [78]. The original PC-PTSD was used to screen for symptoms of re-experiencing, numbing, avoidance, and hyperarousal and had been validated in a Department of VA primary care population [87]. Criterion validity of the PC-PTSD-5 was assessed in a Veteran primary care sample [78] for diagnostic accuracy in comparison to the M.I.N.I. [10]. AUC for the PC-PTSD-5 was 0.94 and the optimally sensitive cut-off score was 3 (sensitivity = 0.95; specificity = 0.85; see Table 2). The PC-PTSD-5 has also been validated in a Korean population of clinical and nonclinical participants [89]. The Korean version showed good internal consistency (α = 0.87) and a cut-off score of 3 resulted in optimal psychometrics (AUC = 0.90; sensitivity = 0.91; specificity = 0.78) [89]. The PC-PTSD-5 appears to be a psychometrically-sound screening questionnaire for PTSD [78].

#### 3.2.7. Short-Form Posttraumatic Checklist-5 (Short-Form PCL-5)

The Short-Form PCL-5 is a 4-item screening questionnaire created by taking items from the original 20-item PCL-5 [16]. The Short-Form PCL-5 was created by examining 160 combinations of between one and eight questions taken from the original PCL-5 [79]. Combinations of the questions were examined with data collected from a large military sample (*n* = 8917). Psychometric properties, along with predictive values of scores on the PCL-5, were examined and a 4-item version with the ability to predict PCL-5 scores of 28 or greater was chosen. The Short-Form PCL-5 was then validated in another large military sample (*n* = 11,728) and showed good psychometric properties (AUC = 0.93; sensitivity = 0.99; specificity = 0.88). The scale has not yet been validated in other languages or cultures.

#### 3.2.8. Short Forms of PTSD Checklist (PCL-C-SF)

Lang and Stein [81] developed the PCL-C-SF, a 2-item, 3-item, 4-item, and 6-item version of the PCL-C [90] to assess for PTSD symptoms in a VA population. The full PCL-C was administered to two samples of primary care VA clients, and sub-samples were contacted for a follow-up interview using the CIDI [91]. The shorter versions of the PCL-C were created using corrected item-total correlations, selecting the items with the highest correlations with the total, as well as items with the highest correlations with the total for each DSM-IV [45] diagnostic criteria cluster. Across the two samples, the 2-item version of the PCL-C resulted in an AUC of 0.73 to 0.88, with a cut-off score of 4 (sensitivity = 0.80 to 0.96; specificity = 0.58 to 0.65) [81]. The 3-item version resulted in an AUC of 0.78 to 0.86, with a cut-off score of 5 (sensitivity = 0.90 to 1.00; specificity = 0.51 to 0.60) [81]. The 4-item version resulted in an AUC of 0.82 to 0.86, with a cut-off score of 8 (sensitivity = 0.80 to 0.83; specificity = 0.68 to 0.70) [81]. The 6-item version resulted in an AUC of 0.84 to 0.89 with a cut-off score of 14 (sensitivity = 0.80 to 0.92; specificity = 0.72 to 0.76; see Table 2); accordingly, Lang and Stein [81] recommend either the 2-item or 6-item version, depending on how much specificity can be sacrificed for a given population (i.e., the more items, the greater specificity). The PCL-C-SF questionnaires have been further validated in VA populations seeking SUD treatment [82]. The 2-item version in the VA SUD sample resulted in an AUC of 0.81, with a cut-off score of 4 (sensitivity = 0.82; specificity = 0.73) [82]. The 3-item version resulted in an AUC of 0.84, with a cut-off score of 6 (sensitivity = 0.78; specificity = 0.72) [82]. The 4-item version resulted in an AUC of 0.85, with a cut-off score of 9 (sensitivity = 0.71; specificity = 0.86) [82]. The 6-item version resulted in an AUC of 0.84, with a cut-off score of 12 (sensitivity = 0.75; specificity = 0.76) [82]. The 4-item version with a cut-off score of 9 is recommended for VA SUD samples [82]. The PCL-C-SF 6-item version has also been used in a PSP sample (paramedics, police, and firefighters) with good internal consistency (α = 0.88) [80]. The PCL-C-SF questionnaires appear to be psychometrically-sound measures for screening PTSD [81].

#### 3.2.9. Short Posttraumatic Stress Disorder Rating Interview (SPRINT)

The SPRINT is an 8-item self-report screening questionnaire for PTSD and related aspects of somatic concerns, stress vulnerability, and functional impairment [83]. The SPRINT consists of 4 items related to the four PTSD symptom clusters (i.e., intrusion, avoidance, numbing, hyperarousal) and 4 items related to somatic distress, distress due to stressful events, interference with work, daily activities, and relationships [83]. Researchers compared the SPRINT to the M.I.N.I. [10] to assess diagnostic accuracy among outpatients participating in a clinical trial of PTSD and a random sample [83]. Cut-off scores of 14–17 in victims of trauma were associated with 96% diagnostic accuracy. Across both samples, a cut-off score of 14 resulted in sensitivity of 0.95 and specificity of 0.96 (see Table 2) [83]. Similar results were found in the Korean version of the SPRINT (a cut-off score of 15 resulted in sensitivity of 0.91 and a specificity of 0.93) and the Korean SPRINT showed good internal consistency (α = 0.86) [92]. The SPRINT appears to be a psychometrically-sound screening questionnaire for PTSD [83]. The conducted search did not identify any studies in which the SPRINT was specifically used with PSP populations.

## 4. Discussion

The reported rates of mental health conditions in Canadian PSP [1] has highlighted the need to find accurate and efficient means of assessing mental disorder symptoms in a timely fashion. Anecdotal feedback has identified the necessity for mental health screening instruments be easy to access, face valid, and concise. The practice of encouraging PSP to complete annual mental health assessments (often called “safeguard assessments”) has become increasingly common, especially as epidemiologic rates of posttraumatic stress injuries and occupational stress injuries in PSP have been increasingly understood [93,94]. Unfortunately, the resource demand for such assessments is often excessive for individual PSP organizations, leading to inconsistent adoption of the safeguard assessment process and limiting its utility. Without regular information regarding the mental health of front-line personnel, PSP organizations may be challenged to implement a consistent and universal mental health strategy, potentially leaving PSP at greater risk of psychological decline [95]. Thus, it is important to develop a mental health screening process that reflects the resource and funding challenges often faced by PSP departments.

Development of a short-form screening battery that assesses for symptoms commonly experienced by PSP could be a means of standardizing mental health assessment for PSP departments, thereby addressing the resource demand problem. Based on the current review, we recommend using the short-form screening battery outlined in Table 3 to screen for symptoms of depression, anxiety, panic disorder, PTSD, and alcohol use. The recommended battery was chosen based on psychometric properties of each scale, validation of the long form or brief form with PSP populations, accessibility of scales for use in clinical and research settings, and the estimated time to complete all measures (~5 min).

### 4.1. Limitations

The review was conducted and the search completed in 2019. Some of the presented screening tools (e.g., PC-PTSD-5, SPRINT) may have since been used or validated with PSP populations. Further, the current study was not a structured review, which means there is a possibility that some uncommon brief screening questionnaires were incidentally excluded in the results.

### 4.2. Conclusions and Directions for Future Research

There are several brief mental disorder screening questionnaires that have been diversely validated in general and clinical populations; nevertheless, there appears to be very few validation studies regarding screening tool use with PSP, and a paucity of questionnaires designed specifically for PSP. Future researchers focused on developing one or more brief mental health disorder screening questionnaires for PSP will also have to carefully balance sensitivity, specificity, relative severity, and symptom cataloguing. Further, the diagnostic accuracy, symptom severity, and symptom profiling available from general or military population data may not generalize to PSP because of their vocational experiences [4,96,97,98]. As such, future researchers should evaluate the generalizability of current screening measures for PSP who appear to be at heightened risk for developing mental disorder symptoms.

## Figures and Tables

**Table 1 ijerph-18-03743-t001:** Features and Empirical Evidence of Brief Mental Disorder Screening Questionnaires (Multiple Disorder Symptom Questionnaires).

Screening Questionnaire	Features	Mental Disorders Screened	Cut-Off Scores	Study/Validation Sample	Psychometric Properties					
					Symptom(s)	Sensitivity	Specificity	AUC	Cronbach’s α	Additional Reliability
90 to 120 Day Post-Deployment Psychological Short Screen	15 items Combination of Yes/No response options and Likert type scales (various response options)	Symptoms related to a potentially traumatic stressor Depression Relationship Problems Alcohol Problems Anger Problems	≥3 Yes responses ≥2 positive responses (More than Half the Days or Nearly Every Day) 2 Yes responses 2 Yes responses Sometimes–Very Often	1578 US soldiers post-deployment and 356 soldiers pre-deployment in the United States [24]	Psychologically Traumatic Stress Depression Relationship Problems Alcohol Problems Anger Problems	0.46 0.50–0.53 0.40–0.82 0.33 0.59	0.97 0.94–0.96 0.89–0.92 0.94 0.91	Not reported	Not reported	Not reported
Anxiety and Depression Detector (ADD)	5 items Yes/No response option	PD PTSD SP GAD MDD	N/A	801 primary care patients in the United States [25]	PD PTSD SP GAD MDD	0.92 0.62 0.69 1.00 0.85	0.74 0.83 0.76 0.56 0.73	Not reported	Not reported	Not reported
Alcohol, Smoking, and Substance Involvement Screening Test (ASSIST)	8 items Yes/No qualifier response option for substance involvement (SI) 5-point Likert type scale for frequency of use (Never–Daily/Almost Daily)	Total Use/Abuse Total Abuse/Dependence Alcohol Use/Abuse Alcohol Abuse/Dependence Cannabis Use/Abuse Cannabis Abuse/Dependence Cocaine Use/Abuse Cocaine Abuse/Dependence Amphetamine Type Stimulant Use/Abuse Amphetamine Type Stimulant Abuse/Dependence Sedative Use/Abuse Sedative Abuse/Dependence Opioid Use/Abuse Opioid Abuse/Dependence	14.5 28.5 5.5 10.5 1.5 10.5 0.5 8.5 0.5 11.5 9.5 10.5 0.5 14.5	1047 participants (697 primary health care patients; 350 patients attending specialized drug treatment facilities) across multiple sites (Australia, Brazil, India, Thailand, United Kingdom, United States, Zimbabwe) [26]	Total Use/Abuse Total Abuse/Dependence Alcohol Use/Abuse Alcohol Abuse/Dependence Cannabis Use/Abuse Cannabis Abuse/Dependence Cocaine Use/Abuse Cocaine Abuse/Dependence Amphetamine Type Stimulant Use/Abuse Amphetamine Type Stimulant Abuse/Dependence Sedative Use/Abuse Sedative Abuse/Dependence Opioid Use/Abuse Opioid Abuse/Dependence	0.80 0.73 0.83 0.67 0.91 0.57 0.92 0.70 0.97 0.72 0.94 0.54 0.94 0.76	0.71 0.66 0.79 0.60 0.90 0.61 0.94 0.77 0.87 0.68 0.91 0.50 0.96 0.65	0.84 0.73 0.87 0.70 0.96 0.62 0.95 0.84 0.96 0.77 0.96 0.45 0.97 0.74	Total SI 0.89 Tobacco Specific SI 0.80 Alcohol Specific SI 0.84 Cannabis Specific SI 0.86 Cocaine SI 0.93 Ampheta-mine Type Stimulant Specific SI 0.94 Sedative Specific SI 0.89 Opioid Specific SI 0.94	Not reported
Composite International Diagnostic-Screener (CID-S)	12 items Yes/No response option	SD PD GAD SP Agoraphobia Specific Phobia Depressive Disorders Hypomania Mania AUD Prescription Drug Disorders Other Illicit Substance Disorders	N/A	Random sample of 1095 participants [27]	SD PD GAD SP Agoraphobia Specific Phobia MDD Mania Alcohol Dependence/Abuse Any Substance Disorder	0.80 1.00 0.91 0.92 0.83 0.92 0.84 1.00 0.79 1.00	0.48 0.47 0.46 0.46 0.48 0.49 0.48 0.46 0.47 0.46	Not reported	Not reported	Test–retest ϰ 0.64–0.92
Composite International Diagnostic Interview Screening Scale (CIDI-SC)	9 preliminary 38 total items Yes/No response option	MDD GAD PD BPD	N/A	3058 patients from primary care offices in the United States [28]	MDE GAD PD BPD	0.80 0.68 0.76 0.74	0.90 0.94 0.94 0.99	0.93 0.88 0.90 0.97	Not reported	Cohen’s ϰ 0.65 0.52 0.62 0.73
Five-Item Mental Health Inventory (MHI-5)	5 items 6-point Likert type scale (All of the Time–None of the Time)	MDD Dysthymia BPD GAD PD Agoraphobia SP Simple Phobia OCD	Not reported	213 individuals newly enrolled in the Harvard Community Health Plan in the United States [29]	Any Affective Disorder Anxiety Disorders MDD	Not reported	Not reported	0.78 0.74 0.89	Not reported	Not reported
			Maximum sensitivity and specificity MDD and Dysthymia 74 GAD 62 PD 70 OCD 78	Random sample of 7076 participants in the Netherlands [30]	Any Mood Disorder MDD Dysthymia BPD Any Anxiety Disorder GAD PD Agoraphobia SP Simple Phobia OCD	0.90 0.74 0.78 1.00	0.80 0.89 0.83 0.78	0.92 0.93 0.91 0.88 0.73 0.90 0.87 0.80 0.77 0.69 0.93	0.83	Not reported
			4-week DSM-IV diagnoses Mood Disorders ≤60 Anxiety Disorders ≤70 SDs ≤60 SUDs ≤70	Random sample of 4075 in Germany [31]	Mood Disorders Anxiety Disorders SDs SUDs	0.83 0.73 0.43 0.67	0.78 0.60 0.78 0.58	0.88 0.71 0.65 0.66	0.74	Not reported
The Kessler Psychological Distress K6 Screening Scale (K6)	6 items 5-point Likert type scale (None of the Time–All of the Time)	Broad screener for serious mental illness (SMI)	Probable Mental Disorder 13 to 24 Moderate Mental Disorder 5	Nationally representative adult samples totaling 41,770 across multiple sites (Brazil, Bulgaria, Colombia, India, Japan, Lebanon. Mexico. New Zealand. Nigeria People’s Republic of China, People’s Republic of China Shenzhen, Romania. South Africa, Ukraine United States) [32]	SMI	Not reported	Not reported	0.76 to 0.86	Not reported	Not reported
			Not reported	78 police officers in Canada [33]	Psychological Distress	Not reported	Not reported	Not reported	0.81	Not reported
			Not reported	459 volunteer firefighters in Australia [34]	Not reported	Not reported	Not reported	Not reported	Not reported	Not reported
The Mental Health Component Scale (MCS-12)	12 items Yes/No response options and varying item length scaled responses per question Score range of 0 to 100 (lower scores reflect poorer mental health).	Broad screener for psychological symptoms and limitations	Depression ≤45 Any Anxiety Disorder ≤50 Any Common Mental Disorder ≤50 Severe Psychological Impairment ≤36	National Survey of Mental Health and Wellbeing 10,504 in Australia [35]	Depression Anxiety Disorders Any common mental disorder	0.87 0.81 0.84	0.83 0.73 0.74	0.92 0.83 0.86	Not reported	Not reported
Mini-International Neuropsychiatric Interview (M.I.N.I.)	21 preliminary items Yes/No response options with optional clinician/researcher rater inquiries	MDD Dysthymia Suicidality Mania PD Agoraphobia SP Specific Phobia OCD GAD Alcohol Dependence Alcohol Abuse Drug Dependence Drug Abuse Psychotic Disorder Anorexia Nervosa PTSD APD	N/A	636 adult sample across the United States and France [10]	MDD Dysthymia Current Mania Lifetime Mania Current PD Lifetime PD Current Agoraphobia Lifetime Agoraphobia Current SP Lifetime SP Current Simple Phobia Lifetime Simple Phobia GAD OCD Current Psychotic Disorder Lifetime Psychotic Disorder Current Alcohol Dependence Current Drug Dependence Lifetime Drug Dependence Anorexia Bulimia PTSD	0.96 0.67 0.82 0.81 0.84 0.88 0.85 0.82 0.81 0.81 0.78 0.70 0.91 0.62 0.84 0.88 0.80 0.45 0.77 0.90 0.92 0.85	0.88 0.99 0.95 0.94 0.93 0.93 0.88 0.92 0.86 0.90 0.90 0.93 0.86 0.98 0.89 0.92 0.95 0.96 0.92 1.00 0.99 0.96	Not reported	Not reported	Interrater/Test–retest ϰ 1.00/0.87 Not reported 0.79/0.35 0.89/0.63 0.92/0.68 0.97/0.79 0.97/0.73 0.92/0.81 0.94/0.65 0.88/0.68 0.88/0.63 0.88/0.52 0.98/0.78 1.00/0.85 0.81/0.77 0.90./0.83 1.00/0.86 0.91/0.96 0.94/0.86 1.00/0.78 1.00/1.00 0.95/0.73
			N/A	100 emergency service personnel (police, firefighters, paramedics) in Australia [36]	Not reported	Not reported	Not reported	Not reported	Not reported	Not reported
My Mood Monitor (M-3) Checklist	27 items 5-point Likert type scale (Not at All–Most of the Time	MDD BPD GAD PD SAD OCD PTSD General Functional Impairment	≥5 ≥2 ≥3 Not reported Not reported Not reported ≥2	647 primary care seeking adults in the United States [37]	MDD BPD Anxiety Disorders PTSD Any Psychiatric Disorder	0.84 0.88 0.82 0.88 0.83	0.80 0.78 0.78 0.76 0.76	Not reported	Not reported	Not reported
Patient Health Questionnaire for Depression and Anxiety (PHQ-4)	4 items 4-point Likert type scale (Not at All–Nearly Every Day)	MDD GAD	Not reported	934 undergraduate students from a Midwestern university in the United States [38]	GAD MDD	Not reported	Not reported	0.84 0.79	GAD-2 0.84 PHQ-2 0.76 PHQ-4 0.81	Test–retest ϰ 0.69–0.81
			≥3 PHQ-2 (probable cases of MDD) ≥3 GAD-2 (probable cases of GAD)	2149 patients from 15 primary care clinics in the United States [39]	GAD PD SAD PTSD GAD PD SAD PTSD	Not reported	Not reported	GAD-2 0.91 0.85 0.83 0.80 PHQ-2 0.81 0.75 0.78 0.77	GAD-2 0.82 PHQ-2 0.75 PHQ-4 0.85	Not reported
				843 police officers in Germany [40]	MDD GAD	Not reported	Not reported	Not reported	0.79	Not reported
Posttraumatic Adjustment Scale (PAS)	10 items 5-point Likert type scale (Not at All–Totally)	PTSD MDE	≥16 ≥4	527 hospitalized injury patients in Australia [41]	PTSD MDE	0.82 0.72	0.84 0.75	0.84 0.73	Not reported	Not reported
Web Screening Questionnaire (WSQ)	15 items Combination of Yes/No response options and Likert type scales (various response options)	MDD GAD PTSD SP PD Agoraphobia Specific Phobia OCD Alcohol Abuse/Dependence	MDD Q1 ≥ 5 and Q2 = 1 GAD Q3 ≥ 2 PD Q4 ≥ 1 PD with Agoraphobia Q4 ≥ 1 and Q5 = 1 Agoraphobia Q5 = 1 Specific Phobia Q6 or Q7 = 1 SP Q8 = 1 and Q9 = 1 PTSD Q10 = 1 or Q11 = 1 OCD Q12 ≥ 1 Alcohol Abuse/Dependence Q 13 ≥ 2 and Q14 ≥ 3 Suicide Q15 = 3 (exclusion)	502 online participants in the Netherlands [42]	MDD GAD SP PD PD with Agoraphobia Agoraphobia Specific Phobia OCD PTSD Alcohol Abuse/Dependence	0.85 0.93 0.72 0.90 0.86 1.00 0.80 0.80 0.83 0.83	0.59 0.45 0.73 0.44 0.77 0.63 0.47 0.69 0.47 0.72	0.72 0.78 0.72 0.76 0.82 0.81 Not reported 0.81 0.65 0.77	Not reported	Not reported
			Not reported	1292 adults (1117 general population, 175 psychiatric outpatients) in the Netherlands [43]	MDD PD PD with Agoraphobia Agoraphobia SP Specific Phobia GAD PTSD OCD Alcohol Abuse/Dependence	0.58 1.00 0.90 0.81 0.79 1.00 0.66 0.79 0.67 0.56	0.94 0.89 0.97 0.95 0.93 0.73 0.90 0.52 0.91 0.93	0.83 0.98 0.99 0.80 0.95 0.93 0.89 0.86 0.82 0.82	Not reported	Not reported

Note. PD = panic disorder; PTSD = posttraumatic stress disorder; SP = social phobia; GAD = generalized anxiety disorder; MDD = major depressive disorder; MDE = major depressive episode; SD = somatoform disorder; AUD = alcohol used disorder; BPD = bipolar disorder; SUD = substance use disorder; APD = antisocial personality disorder; SAD = social anxiety disorder; AUC = area under the receiver operating characteristic (ROC) curve (i.e., the diagnostic power of the questionnaire in discriminating between those with and without the diagnosis of interest); sensitivity = the proportion of true cases detected by the questionnaire; specificity = the proportion of true non-cases correctly classified as non-cases by the questionnaire; RCT = randomized control trial; CBT = cognitive behaviour therapy; ϰ = Kappa co-efficient.

**Table 2 ijerph-18-03743-t002:** Features and Empirical Evidence of Brief Mental Disorder Screening Questionnaires (Single-Disorder Symptom Questionnaires).

Screening Questionnaire	Features	Mental Disorders Screened	Cut-Off Scores	Study/Validation Sample	Psychometric Properties					
					Symptoms	Sensitivity	Specificity	AUC	Cronbach’s α	Additional Reliability
Alcohol Use Disorders Identification Test-Consumption (AUDIT-C)	3 items 5-point Likert type scale of alcohol consumption (Never/0 Drinks–6 or More Times a Week/10 or More Drinks)	Alcohol Abuse/Dependence Heavy Drinking	≥3 for sensitivity ≥4 for specificity	243 VA general medical clinic patients in the United States [68]	Alcohol Abuse/Dependence Heavy Drinking	0.90 0.98	0.45 0.56	0.79 0.89	Not reported	Not reported
			Alcohol misuse ≥4 for males and ≥3 for females	944 firefighters in the United States [71]	Alcohol Abuse/Dependence and Heavy Drinking	Not reported	Not reported	Not reported	0.80	Not reported
			Not reported	266 women firefighters in the United States [72]	Alcohol Abuse/Dependence and Heavy Drinking	Not reported	Not reported	Not reported	0.67	Not reported
			Any AUD ≥7.5 Alcohol Dependence ≥8.5	222 public first responders in South Korea [70]	Any AUD Alcohol Dependence	0.82 0.86	0.80 0.86	Not reported	Not reported	Test–retest ICC 0.91
			Alcohol misuse ≥4 for males and ≥3 for females	184 police officers in the United States [73]	Not reported	Not reported	Not reported	Not reported	Not reported	Not reported
			Not reported	321 career firefighters 204 volunteer firefighters in the United States [74]	Alcohol Abuse/Dependence and Heavy Drinking	Not reported	Not reported	Not reported	0.81	Not reported
Brief Panic Disorder Severity—Self Report (Brief PDSS-SR)	2 items	PD	>3	5103 clients with PD, SAD, or MDD [75]	PD	0.85	0.66	0.82	Not reported	ω_C_ = 0.74
Brief PTSD Screen (Brief PSS)	3 items 4-point Likert type scale (Not at All–Five or More Times per Week/Very Much)	PTSD comorbid with severe mental illness (SMI)	≥3	220 clients with a SMI diagnosis in the United States [76]	PTSD	0.94	0.85	0.96	0.81	Not reported
Panic Disorder Screener (PADIS)	4 items Q1 5-point Likert type scale for number of panic episodes (None–11 or more) Q1 response of 0 does not require completion of remaining items Q 2-4 4-point Likert type scale (Never–All of the Time)	PD	≥4	12,336 young adults in Australia [77]	PD	0.77	0.84	0.87	0.86	Not reported
Patient Health Questionnaire-2	2 items 4-point Likert type scale (Not at All–Nearly Every Day)	MDD	≥3	6000 patients in primary care and obstetrics-gynecology clinics in the United States [63]	MDD	0.83	0.92	0.93	Not reported	Not reported
			Not reported	184 police officers in the United States [73]	Not reported	Not reported	Not reported	Not reported	Not reported	Not reported
Primary Care Posttraumatic Stress Disorder screen (PC-PTSD-5)	5 items Yes/No response options	PTSD	≥3 Yes	398 VA primary care clients in the United States [78]	PTSD	0.95	0.85	0.94	Not reported	Not reported
Short-Form PCL-5	4 items 5-point Likert type scale (Not at All–Extremely)	PTSD	Not reported	8917 active duty military to select items; 11728 for validation [79]	PTSD	0.99	0.88	0.93	Not reported	Not reported
Short Forms of PTSD Checklist (PCL-C-SF)	2 items to 6 items 5-point Likert type scale (Not at All–Extremely)	PTSD	6 items	986 first responders (police, firefighters, paramedics) [80]	PTSD	Not reported	Not reported	Not reported	0.88	Not reported
			2 items ≥4 6 items ≥14	221 VA primary care female patients 154 VA primary care patients in the United States [81]	PTSD	2-item version 0.80–0.96 6-item version 0.80–0.92	2-item version 0.58–0.65 6-item version 0.72–0.86	2-item version 0.73–0.88 6-item version 0.84–0.89	Not reported	Not reported
			4 items ≥9	Primary care VA patients seeking treatment in SUD specialty treatment (158) and general mental health treatment (242) [82]	PTSD	2-item version 0.72 4-item version 0.65 6-item version 0.75	2-item version 0.73 4-item version 0.90 6-item version 0.78	2-item version 0.81 4-item version 0.85 6-item version 0.84	Not reported	Not reported
Short Posttraumatic Stress Disorder Rating Interview (SPRINT)	8 items 5-point Likert type scale (Not at All–Very Much)	PTSD and related symptoms	14 to 17	83 outpatients in clinical trial of PTSD and 630 randomly selected participants in the United States [83]	PTSD and related symptoms	Total sample 0.95 Clinical sample 0.62	Total sample 0.96 Clinical sample 0.94	Not reported	0.78–0.88	Test–retest ICC 0.78 Interrater ICC 1.00

Note. PD = panic disorder; PTSD = posttraumatic stress disorder; GAD = generalized anxiety disorder; MDD = major depressive disorder; MDE = major depressive episode; SD = somatoform disorder; AUD = alcohol used disorder; SUD = substance use disorder; SAD = social anxiety disorder; VA = veterans affairs; AUC = area under the receiver operating characteristic (ROC) curve (i.e., the diagnostic power of the questionnaire in discriminating between those with and without the diagnosis of interest); sensitivity = the proportion of true cases detected by the questionnaire; specificity = the proportion of true non-cases correctly classified as non-cases by the questionnaire; ICC = intraclass correlation coefficient.

**Table 3 ijerph-18-03743-t003:** Recommended short-form screening battery for PSP.

Domain(s)	Recommended Measure	Number of Items
Depression and Anxiety	PHQ-4 [39]	4
Panic Disorder	Brief PDSS-SR [75]	2
PTSD	Short-Form PCL-5 [79]	4
Alcohol Use Disorder	AUDIT-C [68]	3
Estimated time to complete		5 min

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
