# Peer review of "Brief Mental Health Disorder Screening Questionnaires and Use with Public Safety Personnel: A Review"

_ijerph, 2021, doi:10.3390/ijerph18073743_

Round 1

Reviewer 1 Report

  1. The subject of this research is the use of medical testing, which is an efficient method to guide doctors to understand the patient's condition briefly.
  2. Introduction section and Motivations  – Authors should add answer the questions:
    • Why you use the methods in this study?
    • What are limitations of existing approaches that exists in literature?
    • What are benefits of this questionnaires?
    • What is novelty of this study?
  3. Literature Review- Authors need to improve this section.
    • There is no complete literature review description for the selection of questionnaire items.
    • No objective preliminary screening process for the selection of questionnaire items.
  4. The content of the table is important, but it is not easy to read because of the content cut.

Author Response

Reviewer 1 Feedback

We very much appreciate this reviewer's feedback. Please find our responses italicized below!

The subject of this research is the use of medical testing, which is an efficient method to guide doctors to understand the patient’s condition briefly.

We very much appreciate the reviewer’s support.

  1. Introduction and Motivations Section – Authors should add answer the questions:
    • Why you use the methods in this study?
    • What are the limitations of existing approaches that exists in literature?
    • What are benefits of this questionnaires?
    • What is novelty of this study?

Thank you for your comment. We have made adjustments to the introduction section, including adding an objective section, in order to address this comment. These changes can be seen with via track changes throughout the revised submission.

  1. Literature Review – Authors need to improve this section.
    • There is no complete literature review description for the selection of questionnaire items.
    • No objective preliminary screening process for the selection of questionnaire items.

Thank you for your comment. We attempted to address this comment by making changes to the Methods section, including an entire section entitled “Search Strategy.” The added section is as follows:

“2.1 Search Strategy

The review was based on search results from several databases; specifically, PsycARTICLES (Ovid), PsycInfo (Ovid), Web of Science, and Google Scholar. An initial search was conducted using the following search terms: brief psychology screening instruments, mental disorder screening, short form of mental disorder measures, psychological symptom questionnaires, short symptom measures, depression symptom measures, anxiety symptom measures, alcohol symptom measures, substance use symptom measures, and post-trauma symptom measures. Variations and combinations of the search terms were used to follow-up on preliminary searches and identify additional questionnaires. Publications written in a language other than English, as well as dissertations and books in any language, were excluded. Brief tools that did not screen for at least one of the five major disorders (i.e., alcohol use disorder, anxiety, depression, panic disorder, and PTSD) were also not included. Validation papers identified for each brief screening questionnaire were located using Google Scholar. A list of brief screening questionnaires that were validated in clinical or general populations was then compiled.

A subsequent search was conducted to explore the use of the identified questionnaires among PSP. Databases used for the subsequent search included PsycInfo (Ovid) and Google Scholar. Abbreviations for previously identified brief screening questionnaires were used as PsycInfo search terms. The abbreviations included were as follows: 90 to 120 Day Post-Deployment Psychological Short Screen, ADD, ASSIST, CID-S, CIDI-SC, MHI-5, K6, MCS-12, M.I.N.I., M-3, PHQ-4, PAS, WSQ, AUDIT-C, Brief PSS, PADIS, PHQ-2, PC-PTSD-5, PCL-C-SF, and SPRINT. PSP job titles (i.e., police, firefighter, paramedic, corrections officer, public safety personnel, and first responder) were paired with each of the brief screening questionnaire abbreviations in order to identify studies using the screening questionnaires with PSP participants. Finally, a search was conducted to identify associated extant research specifically validating each tool for use with PSP.”

  1. The content of the table is important, but not easy to read because of the content cut.

We agree and have removed some unnecessary data, as well as combined the referencing and study description into one column. We hope these changes will make the table more readable.

Thank you again for your valuable feedback. We hope that we have addressed all of your comments adequately.

Reviewer 2 Report

Dear colleagues, I hope this message find you well.

Thank you for giving me the opportunity of reading the work “Brief Mental Health Disorder Screening Questionnaires and Use with Public Safety Personnel: A Review, it has been a very big pleasure to collaborate reviewing this manuscript. The topic of this paper is very interesting and it seems necessary to delve it. However, there are several questions to improve before to publish it. I would suggest some changes: 

Abstract:

  • It is advisable to reduce the contextual explanation. Moreover, you should name the main findings.
  • I suggest to add more keywords

Introduction:

  • The structure of the introduction is not clear. I recommended to divide it into subsections. For example, I recommend to create a specific subsection where describe objectives specifically.
  • The numbering of the upper right margin is incorrect
  • In relation to page 2, I recommend creating a table to synthesize all the exposed instruments.
  • Table 1 is too long, it is difficult to interpret it. Find a way to reduce the size. Organizing your content in a different way could be a solution.

Discussion

  • It is necessary to describe study limitations and practical implications. You must defend the main contributions and finding more deeply.

Conclusions

  • Future research should be in the previous section.

References

Please, you must follow the standards proposed by MDPI. For example:

  • References in text must follow a sequentially numeration.
  • Name of the journals abbreviated and in italics.
  • Add DOIs when it will be possible.

Author Response

We very much appreciate this reviewer's feedback and support. We have addressed their comments as outline below in italics.

Dear colleagues, I hope this message find you well. Thank you for giving me the opportunity of reading the work “Brief Mental Health Disorder Screening Questionnaires and Use with Public Safety Personnel: A Review”, it has been a very big pleasure to collaborate reviewing this manuscript. The topic of this paper is very interesting and it seems necessary to delve it. However, there are several questions to improve before to publish it.

We very much appreciate the reviewer’s support.

  1. Abstract:
    • It is advisable to reduce the contextual explanation. Moreover, you should name the main findings.

Thank you for your comment. We have made the following changes to the abstract to in an attempt to address this comment.

Abstract: Brief mental health disorder screening questionnaires (SQs) are used by psychiatrists, physicians, researchers, psychologists, and other mental health professionals and may provide an efficient method to guide clinicians to query symptom areas requiring further assessment. Annual screening has been used to help identify military personnel who may need help. Nearly half (44.5%) of Canadian public safety personnel (PSP) screen positive for one or more mental health disorder(s); as such, regular mental health screenings for PSP may be a valuable way to support mental health. The following review was conducted to 1) identify existing brief mental health disorder SQs; 2) review empirical evidence of the validity of identified SQs; 3) identify SQs validated within PSP populations; and 4) recommend appropriately validated brief screening questionnaires for five common mental health disorders (i.e., generalized anxiety disorder [GAD], major depressive depression [MDD], panic disorder, posttraumatic stress disorder, alcohol use disorder) . After reviewing the psychometric properties of the identified brief screening questionnaires, we recommend the following four brief screening tools for use with PSP: the Patient Health Questionnaire-4 (screening for MDD and GAD), the Brief Panic Disorder Symptom Screen–Self-Report, the Short Form Posttraumatic Checklist-5, and the Alcohol Use Disorders Identification Test-Consumption.”

  • I suggest to add more keywords.

Thank you for the suggestion. We have added the following keywords: questionnaires; MDD; GAD; panic disorder; PTSD; alcohol use disorder. If the reviewer or editor have other suggestions, we would be delighted to include them.

  1. Introduction:
    • The structure of the introduction is not clear. I recommended to divide it into subsections. For example, I recommend to create a specific subsection where describe objectives specifically.

Thank you for this suggestion. The following objective section has been added in order to clearly outline the purpose of the study.

“1.2 Objectives

The current study was designed to address the following objectives: 1) identify existing brief mental health disorder screening questionnaires; 2) review empirical evidence of the validity of identified screening questionnaires; 3) identify screening questionnaires validated within PSP populations; and 4) provide recommendations for appropriate and validated brief screening questionnaires for PSP.”

  • The numbering of the upper right margin is incorrect.

Thank you for catching this error. The numbering has been corrected.

  • In relation to page 2, I recommend creating a table to synthesize all the exposed instruments.

We agree that the additional table may be helpful for some readers. However, based on other reviewer feedback, we are concerned that the additional table may be redundant. We are willing to provide the table, pending editorial request.

  • Table 1 is too long, it is difficult to interpret it. Find a way to reduce the size. Organizing your content in a different way could be a solution.

We agree and have removed some unnecessary data, as well as making some changes to the way the content was organized. We hope these changes will make the table more readable.

  1. Discussion:
    • It is necessary to describe study limitation and practical implications. You must defend the main contributions and finding more deeply.

Thank you for the comment. We have outlined the contributions of our findings and our paper in the introduction, discussion, and conclusion. We are happy to elaborate further with specific recommendations from the reviewer or editor.

We have also added the following limitations section in an effort to address this comment.

“4.0 Limitations

The review was conducted with the search completed in 2019. Some of the presented screening tools (e.g., PC-PTSD-5, SPRINT) may have since been used or validated with PSP populations. Further, the current study was not a structured review, which means there is a possibility that some uncommon brief screening questionnaires were incidentally excluded in the results.”

  1. Conclusions:
    • Future research should be in the previous section.

Thank you for your comment. It appears that our headings and subheadings were not laid out clearly enough. We have adjusted the headings so it is now clear that the “Conclusions and Directions for Future Research” section is a part of the “Discussion” section. For example, we changed the “Discussion” heading to now read “4.0 Discussion” and the “Conclusions and Directions for Future Research” heading to now read “4.2 Conclusions and Directions for Future Research.”

  1. References:
    • Please, you must follow the standards proposed by MDPI. For example:
      • References in text must follow a sequentially numeration.
      • Name of the journals abbreviated and in italics.
      • Add DOIs when it will be possible.

Thank you for your comment. We have made the appropriate referencing format changes to match the MDPI requirements.

Thank you again for your valuable feedback. We hope we have successfully addressed all of your comments.

Reviewer 3 Report

This is a well written paper describing tests used to assess mental health in public service personnel.  This paper describes a number of tests that have been used, their validity and reliability and suggest a number of tests, that take less time to complete, that can be used to diagnose mental health issues in this group of workers.  The reviewer has attached the manuscript with some minor editorial that may help clarify a few of the ideas the authors are trying to convey.

Author Response

Thank you for your valuable feedback. We have addressed the comments as outline below in italics.

This is a well written paper describing tests used to assess mental health in public service personnel.  This paper describes a number of tests that have been used, their validity and reliability and suggest a number of tests, that take less time to complete, that can be used to diagnose mental health issues in this group of workers. 

                                 We very much appreciate the reviewer’s support.

  1. The reviewer has attached the manuscript with some minor editorial that may help clarify a few of the ideas the authors are trying to convey.

We have addressed the reviewer’s editorial changes from the attached manuscript. In the sentence “Screening questionnaires cannot provide a definitive diagnosis of a mental health disorder but are designed to help clinicians more efficiently identify symptoms indicative of a mental health disorder requiring further assessment [9] and to help increase diagnostic consistency [10],” we did indeed mean to use the word consistency.

Thank you again for your valuable feedback. We hope that we have successfully addressed your comments. 

Reviewer 4 Report

Major disorders like anxiety, depression and PTSD are prevalent among high stress occupations; and it’s paramount to address, assess and mitigate this issue by identifying the conditions in early stages , making mental health services accessible in terms of short form screening tools.

In my opinion this manuscript was is very well written and very informative. I did not find any major issues; however, I do have some minor suggestions and comments for the authors to consider:

  • Page 4: Please describe what M.I.N.I. is before using its acronym
  • Page 4-5: the ADD instrument. Not to be confused with Attention Deficit Disorder. Maybe mention that in the text?
  • Table 1: It looks a bit involved and complicated; hard to read and identify necessary information due to some of the columns being too narrow. I wonder if its possible to split information somehow or make the font smaller (just a suggestion).
  • I would advise inserting Table 1 after the sections introducing all the questionnaires reported in Table 1.
  • Table 2: Same comments for Table 2.
  • Numerous acronyms are used throughout the text. The authors should make sure they introduce a full description of the instrument prior to using its acronym.

The review of various short form mental health instruments in this manuscript was performed in detailed and descriptive manner, which will be appreciated by the mental health research community.

Author Response

Thank you so much for your valuable feedback. We have outlined our responses below in italics. 

Major disorders like anxiety, depression and PTSD are prevalent among high stress occupations; and it’s paramount to address, assess and mitigate this issue by identifying the conditions in early stages , making mental health services accessible in terms of short form screening tools.

In my opinion this manuscript was is very well written and very informative. I did not find any major issues.

                                 We very much appreciate the reviewer’s support.

  1. I do have some minor suggestions and comments for the authors to consider:
    • Page 4: Please describe what M.I.N.I is before using its acronym.

         Thank you for catching this error. It has been corrected.

  • Page 4-5: the ADD instrument. Not to be confused with Attention Deficit Disorder. May mention that in the text?

Thank you for this suggestion. Although we understand that referring to the ADD instrument as just “ADD” could be confused with the ADD disorder, the authors of the original ADD validation paper refer to their instrument as the ADD (see Means-Christensen et al., 2006). We have decided to use their suggested language when referring to the ADD in our paper.

  • Table 1 & 2 - Readability: It looks a bit involved and complicated; hard to read and identify necessary information due to some of the columns being too narrow. I wonder if it’s possible to split information somehow or make the font smaller (just a suggestion).

We agree and have removed some unnecessary data, as well as making some changes to the way the content was organized. We hope these changes will make the table more readable.

  • Table 1 & 2 - Insertion: I would advise inserting Table 1 after the sections introducing all the questionnaires reported in Table 1.

We very much appreciate your suggestion; however, the IJERPH has requested that tables be inserted into the text immediately after the first mention of the table.

  • Numerous acronyms are used throughout the text. The authors should make sure they introduce a full description of the instrument prior to using its acronym.

Thank you for your comment. We have re-read the paper and ensured that all acronyms were first described prior to use. If the reviewer or editor feels necessary, we are very happy to provide an abbreviation table as a supplementary material.

  1. The review of various short form mental health instruments in this manuscript was performed in detailed and descriptive manner, which will be appreciated by the mental health research community.

Thank you so much for your support and kind words. We hope that we have successfully addressed your comments.